



# Uptake of Organic Vapors and Nitric Acid on Atmospheric Freshly Nucleated Particles

Yosef Knattrup and Jonas Elm

Department of Chemistry, Aarhus University, Langelandsgade 140, 8000 Aarhus C, Denmark

**Correspondence:** Jonas Elm (jelm@chem.au.dk)

**Abstract.** Sulfuric acid, ammonia, and amines are believed to be key contributors to the initial steps in new particle formation in the atmosphere. However, other compounds such as organic compounds or nitric acid are believed to be important for further growth at larger sizes. In this study, we investigate the potential uptake of first-generation oxidation products from $\alpha$-pinene (pinic and pinonic acid), and isoprene (trans-$\beta$-IEPOX, $\beta$4-ISPOOH, and $\beta$1-ISOPOH), a potential highly oxidized molecule

(HOM), formic acid, and nitric acid. The uptake is probed onto $(SA)_{10}(base)_{10}$ freshly nucleated particles (FNPs), where SA denotes sulfuric acid and the bases are either ammonia (AM), methylamine (MA), dimethylamine (DMA), or trimethylamine (TMA). The addition free energies were calculated at the $\omega$B97X-D3BJ/6-311++G(3df,3pd)//B97-3c level of theory. We find favorable addition free energies of $-8$ to $-10$ kcal/mol for the HOM, pinic acid, and pinonic acid on the less sterically hindered $(SA)_{10}(AM)_{10}$ and $(SA)_{10}(MA)_{10}$ FNPs. This suggests that isoprene oxidation products do not contribute to the early growth

of FNPs, but the $\alpha$-pinene products do, in accordance with their expected volatilities.

Calculating the second addition of a pinic acid molecule or pinonic acid molecule on the $(SA)_{10}(AM)_{10}$ FNPs, we find that pinic acid maintains its large addition free energy decrease due to its two carboxylic acid groups interacting with the other monomer as well as the FNP. The pinonic acid addition free energy drops to $-3.9$ kcal/mol due to the weak interactions between the FNP and its carbonyl group and the lack of monomer–monomer interactions. The high potential for pinic addition

is confirmed by calculating the addition free energy at realistic atmospheric conditions. This means that pinic acid has the potential for organic growth on $\sim$2 nm FNPs, implying that other dicarboxylic acids could potentially also aid in the early growth.

## 1 Introduction

Clouds play a significant role in shaping the global climate and their formation begins when water condenses onto cloud

condensation nuclei (CCN). (Boucher and Lohmann, 1995) Aerosols can act as CCN delivering the required surface area for water to condense on when they reach sizes of roughly 50–100 nm in diameter. (Boucher and Lohmann, 1995) The cloud–aerosol interaction is, as of the 5th IPCC assessment report, the largest cause of uncertainty in modern radiative forcing modeling. (Canadell et al., 2021) Aerosols form through two main pathways: either as primary particles, directly emitted into the atmosphere, or as secondary particles, which form through the clustering of gas vapors. (Kulmala et al., 2013) Around

50% of CCN are believed to originate from the secondary process (Boucher and Lohmann, 1995; Merikanto et al., 2009)



denoted new particle formation (NPF). As an example, Zhao et al. (2024) simulated the spatial distribution of CCN at the surface level and found the fraction of CCN from NPF to be around 30–40% over mainland Europe and up to ≈ 60% over eastern United States.

The substantial uncertainty in the climate models is linked with the unidentified mechanisms for the NPF pathways and the early growth behavior of freshly nucleated particles (FNPs). (Canadell et al., 2021) For instance, Tröstl et al. (2016) found a factor two of difference in predicted CCN number concentration when altering the 1.7–3.0 nm particles growth mechanisms in their global model. It is therefore important to elucidate the initial growth mechanisms. Wang et al. (2020) found experimentally that vapors of ammonia (AM) and nitric acid (NA) could condense onto freshly nucleated particles (FNPs) at 278.5 K but temperatures below 258.15 K were required for nucleation of NA and AM. They extended this study (Wang et al., 2022) to include sulfuric acid (SA) and found that NA, SA, and AM could form particles in conditions similar to the upper free troposphere. Stolzenburg et al. (2018) studied the influence of organics in early particle growth in the CLOUD chamber and found rapid growth from organics in temperature ranges of 248.15–298.15 K.

Experimentally the main focus has been on measuring particles larger than 2 nm in diameters as particles below this size are challenging to measure accurately. Sub 2 nm charged cluster compositions have been measured by mass spectrometry techniques (Jokinen et al., 2012), however, it is uncertain if the measured clusters underwent fragmentation inside the instrument or if the charging of neutral clusters changes the composition during the measurement. (Zapadinsky et al., 2019; Passananti et al., 2019) This positions quantum chemical methods as a key tool for exploring the formation of small clusters and their early growth processes.

The initial clustering process is believed to primarily be driven by sulfuric acid (SA) stabilized by ammonia (AM) (Kulmala et al., 2013; Kirkby et al., 2011; Schobesberger et al., 2013; Weber et al., 1996; Elm, 2021a; Dunne et al., 2016; Kubečka et al., 2023b) or strong amines such as methylamine (MA) (Kurtén et al., 2008; Nadykto et al., 2011, 2015, 2014; Jen et al., 2014; Glasoe et al., 2015; Kubečka et al., 2023b), dimethylamine (DMA) (Kurtén et al., 2008; Loukonen et al., 2010; Nadykto et al., 2011, 2015, 2014; Jen et al., 2014; Glasoe et al., 2015; Almeida et al., 2013; Elm, 2021a; Kubečka et al., 2023b) and trimethylamine (TMA) (Elm, 2021a; Kurtén et al., 2008; Nadykto et al., 2011; Jen et al., 2014; Nadykto et al., 2015; Glasoe et al., 2015; Kubečka et al., 2023b). Nitric acid (NA) (Wang et al., 2020; Liu et al., 2021, 2018; Kumar et al., 2018; Ling et al., 2017; Wang et al., 2022; Nguyen et al., 1997; Longsworth et al., 2023; Knattrup et al., 2023; Knattrup and Elm, 2022; Bready et al., 2022; Qiao et al., 2024) and formic acid (FA) (Bready et al., 2022; Knattrup et al., 2023; Ayoubi et al., 2023; Zhang et al., 2022, 2018; Harold et al., 2022; Nadykto and Yu, 2007) have also been shown to be involved in the initial clustering. Organics have not been definitively proven to take part in the initial clustering process but they are known to be important for the growth. (Kulmala et al., 2013; Elm et al., 2020, 2023; Engsvang et al., 2023b)

While much of the focus has been on the initial clustering process (up to 8 monomers) (Elm et al., 2020, 2023; Engsvang et al., 2023b), Engsvang et al. (Engsvang et al., 2023a; Engsvang and Elm, 2022) and Wu et al. (Wu et al., 2023, 2024) has pushed the studies up to large particles of up to 30–60 monomers, reaching geometric diameters of up to 2 nm, where the clusters start exhibiting more particle-like properties. Wu et al. (2024) defined the cluster-to-particle transition point as the point where the free energy per monomer starts leveling off, thereby resembling "bulk" thermodynamics and where





"solvated" monomers appear in the cluster structures. They found this point to be around 10 acid-base pairs for the SA–AM/MA/DMA/TMA clusters and defined the clusters at and beyond this point as FNPs. Previously, DePalma et al. (2015) studied the uptake of pinic and pinonaldehyde on a $(SA)_4(AM)_4$ cluster at the AM1 level of theory. However, the early growth of realistic sizes ~2 nm FNPs has yet to be studied.

65    In this paper, we study the uptake of NA and common organics on the $(SA)_{10}(AM/MA/DMA/TMA)_{10}$ FNPs using quantum chemical methods. We investigate the first-generation oxidation products of $\alpha$-pinene, isoprene (Nozière et al., 2015) and a potential highly oxidized molecule (HOM) of $\alpha$-pinene (Kurtén et al., 2016) (suggested by COSMO-RS calculations). The studied systems are displayed in Figure 1.

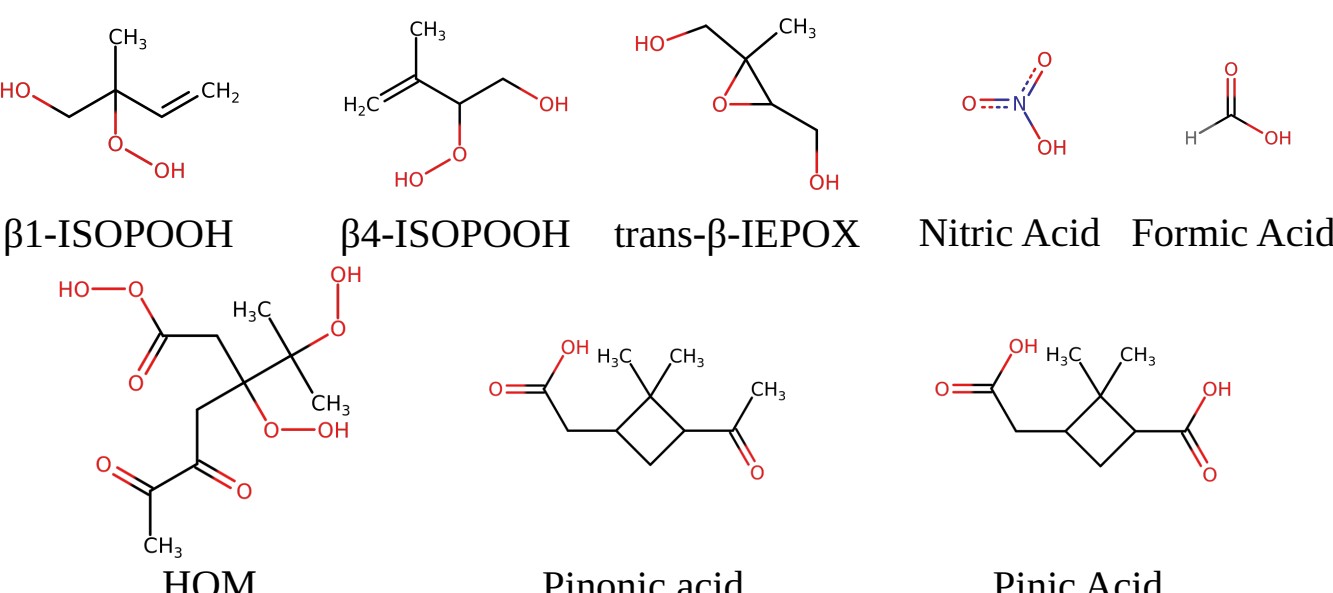

**Figure 1.** The molecular structure of the studied monomers. Pinonic and pinic acid are first-generation oxidation products from $\alpha$-pinene. trans-$\beta$-IEPOX, $\beta$4-ISOPOOH, and $\beta$1-ISOPOOH are first-generation oxidation product from isoprene.

70  **2  Methodology**

**2.1  Computational Details**

The B97-3c (Brandenburg et al., 2018) and $\omega$B97X-D3BJ (Najibi and Goerigk, 2018)/6-311++G(3df,3pd) (Ditchfield et al., 1971) calculations were performed in ORCA 5.0.4 (Neese, 2012; Neese et al., 2020; Neese, 2022). The xTB 6.4.0 (Bannwarth et al., 2021) program was used for the GFN1-xTB[re-par] calculations. GFN1-xTB[re-par] is a parameterization of GFN1-xTB
75  (Grimme et al., 2017) using the methodology suggested by Knattrup et al. (2024). This specific parameterization was done by Wu et al. (2024) where a linear combination of the binding energy and gradient errors were minimized on a set of B97-3c FNPs. Configurational sampling was performed using ABCluster 3.2 (Zhang and Dolg, 2015, 2016; Zhang, 2022) and CREST



2.12 (Grimme, 2019; Pracht et al., 2020) in noncovalent interaction mode. The entire computational workflow and subsequent data handling was performed using the JKCS 2.1 suite of programs. (Kubečka et al., 2023a). All the data is freely available in the atmospheric cluster database. (Kubečka et al., 2023a; Elm, 2019) See the supporting information for more information.

## 2.2 Thermochemistry

We define the binding free energy $\Delta G_{\text{bind}}$ as the free energy change between the combined cluster ($G_{\text{cluster}}$) and the free energy ($G_{\text{monomer}}$) from the $n$ monomers,

$$\Delta G_{\text{bind}} = G_{\text{cluster}} - \sum_{i}^{n} G_{\text{monomer}}^{i}. \tag{1}$$

To get more accurate free energies, we performed a higher level single point correction on top of the DFT geometries. This is denoted with the sp//geo notation where "sp" is the single point energy method used to calculate the electronic energy ($E^{\text{sp}}$) and "geo" is the method used to calculate the thermal correction to the free energy ($G_{\text{thermal}}^{\text{geo}}$) from the optimized structure and vibrational frequencies. The thermal correction term includes everything except the electronic energy:

$$G = E^{\text{sp}} + G_{\text{thermal}}^{\text{geo}}. \tag{2}$$

The addition free energy is the binding free energy change when a monomer is added to the cluster:

$$\Delta G_{\text{add}} = \Delta G_{\text{bind}}^{\text{cluster+uptake}} - \Delta G_{\text{bind}}^{\text{cluster}}. \tag{3}$$

We employ the quasi-harmonic approximation (Grimme, 2012) (as standard in ORCA), where vibrational frequencies below $100 \text{ cm}^{-1}$ are treated as free rotor contributions to the entropy.

$$S_{\text{rot,qh}} = \frac{1}{2}R + R\ln\left[\left(\frac{8\pi^3 I' k_{\text{B}} T}{h^2}\right)^{1/2}\right], \tag{4}$$

where $k_{\text{B}}$ is the Boltzmann constant, $T$ the temperature, $I'$ the effective moment of inertia, $R$ the gas constant, and $h$ is the Planck constant. The output of quantum chemical programs are, for the most part, at standard conditions ($p = 1$ atm, $T = 298.5$ K). If the binding free energy is needed at other conditions, Halonen (2022) derived the following equation for the binding free energy at the given temperature and concentration,

$$\Delta G_{\text{bind}}(\boldsymbol{p}) = \Delta G_{\text{bind}}^{\text{ref}} - RT\left(1 - \frac{1}{n}\right)\sum_{i}^{n}\ln\left(\frac{p_i}{p_{\text{ref}}}\right), \tag{5}$$

where $p_i$ is the partial pressure (concentration) of monomer $i$, $n$ is the number of monomers, and $p_i$ is the reference pressure the reference binding free energy was calculated at. This equation correctly satisfies self-consistency for multi-component clusters (the monomers will have a free energy of zero) and the law of mass action. Combining Equation 5 with Equation 3 yields the addition free energy at the given conditions,

$$\Delta G_{\text{add}}(\boldsymbol{p}) = \Delta G_{\text{add}}^{\text{ref}} - \frac{RT}{n(n-1)}\left(\sum_{i}^{n-1}\ln\left(\frac{p_i}{p_{\text{ref}}}\right) + (n-1)^2\ln\left(\frac{p_{\text{add}}}{p_{\text{ref}}}\right)\right), \tag{6}$$

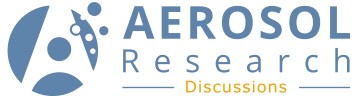

now $n$ is the total number of monomers in the largest cluster, $\Delta G_{\text{add}}^{\text{ref}}$ is the reference addition free energy, the sum runs over all monomers in the clusters before the addition, and $p_{\text{add}}$ is the pressure (concentration) of the added monomer. The temperature at which the addition is spontaneous at the given concentrations can be found by setting Equation (6) to zero and solving for the temperature,

$$T_{\text{spont}} = \Delta G_{\text{add}}^{\text{ref}} \frac{n\,(n-1)}{R} \left( \sum_{i}^{n-1} \ln\left( \frac{p_i}{p_{\text{ref}}} \right) + (n-1)^2 \ln\left( \frac{p_{\text{add}}}{p_{\text{ref}}} \right) \right)^{-1}. \tag{7}$$

## 2.3 Configurational Sampling Workflow

We employed the configurational workflow for FNPs as suggested by Wu et al. (2023, 2024).

$$\text{ABC} \xrightarrow{N=10,000} \text{xTB}^{\text{OPT}} \xrightarrow{N=10,000} \text{B97-3c}^{\text{SP}} \xrightarrow[\text{filter}]{N=1,000} \text{B97-3c}^{\text{PART OPT}} \xrightarrow[\text{filter}]{N=100} \text{B97-3c}^{\text{FULL OPT}} \tag{8}$$

The workflow entails 10 parallel runs of ABCluster (Zhang and Dolg, 2015, 2016) with $SN = 1280$, $gen = 320$ and $sc = 4$. We used matching ionic monomers leading to an overall electronic neutral cluster. We choose to have ionic inorganic monomers and neutral organic monomers. We chose this because SA is a stronger acid compared to the monomers. All the structures from ABCluster are subsequently optimized with GFN1-xTB$^{\text{re-par}}$ and have a single point energy calculated at the B97-3c level of theory. The 1000 structures lowest in electronic energy are then partially optimized with B97-3c for 40 iterations. The 100 structures lowest in electronic energy are then fully optimized and finally, a vibrational frequency calculation is performed.

The lowest free energy is used as the input structure for CREST using GFN1-xTB$^{\text{re-par}}$, as suggested by Knattrup et al. (2024), and the 100 structures lowest in electronic energy are then fully optimized followed by a vibrational frequency calculation.

$$\text{CREST(GFN1-xTB}^{\text{re-par}}) \xrightarrow{N=100} \text{B97-3c}^{\text{FULL OPT}} \tag{9}$$

The larger clusters suffered from memory issues and SCF convergence problems causing a low number of fully optimized structures using our automated workflow. We then restarted all 200 cluster calculations with increased memory and `VeryTightSCF`, and redid the CREST workflow, resulting in some cluster systems having over 200 configurations. The organic monomers were sampled using CREST. The ABCluster input was generated from the lowest energy conformer using topgen (Zhang, 2022; Zhang and Dolg, 2015, 2016) with charges from an MP2/6-31++G** calculation in Gaussian 16.

### 2.4 Single Point Refinement

Engsvang et al. (2023a) found that the B97-3c single point energies gave quite erroneous addition free energies for FNPs compared to the DLPNO-CCSD(T$_0$)/aug-cc-pVTZ reference calculations. They found that $\omega$B97X-D3BJ/6-311++G(3df,3pd) had excellent error cancellation for the addition free energies yielding errors below 1 kcal/mol. Hasan et al. (2024) extended the benchmark by including single point calculations with $\omega$B97X-D3BJ and the augmented def2 basis sets. However, none managed to beat the combination found by Engsvang et al. (2023a) We, therefore, performed $\omega$B97X-D3BJ/6-311++G(3df,3pd) single point energy calculations on the 10 lowest free energy configurations. The SCF calculations had trouble converging due to linear dependencies in the basis, therefore we had to increase the `scfthres` parameter to 1e-6. This change slightly



increases the energy (max of 0.6 kcal/mol for the 10 acid 10 base systems), however, it is systematic for the given system size and should partly cancel out for the binding addition free energies.

It should be noted that the $\omega$B97X-D3BJ/6-311++G(3df,3pd) single point energies slightly overbind (roughly 4 kcal/mol) compared to the DLPNO-CCSD(T$_0$)/aug-cc-pVTZ reference level by Engsvang et al. (2023a) However, all the energies are roughly shifted the same, leading to less erroneous addition free energies.

## 3 Results and Discussion

### 3.1 Addition Free Energies

To probe the potential for the (SA)$_{10}$(AM/MA/DMA/TMA)$_{10}$ FNPs to uptake monomers (pinic and pinonic acid, trans-$\beta$-IEPOX, $\beta$4-ISPOOH, $\beta$1-ISOPOH), HOM, FA, and NA, we calculated the addition free energies as defined in equation (3).

The reference FNPs without the organics are taken from Wu et al. (2024). They found that for realistic atmospheric conditions

([SA] $= 10^6$ molecules cm$^{-3}$, AM = (10 ppt, 10 ppb), MA/DMA/TMA = (1, 10 ppt)), the formation of the SA–AM and SA–MA FNPs go through nucleation barriers but are stable after the barrier. Furthermore, the formation of the SA–DMA FNP is found to be entirely barrierless but the formation of the SA–TMA FNP is unfavorable (due to steric hindrance) as the binding free energy is positive for the studied conditions.

Figure 2 presents the calculated addition free energies, at the $\omega$B97X-D3BJ/6-311++G(3df,3pd)//B97-3c level of theory. In

the following sections we will discuss the trends for each FNP type.







**Figure 2.** The addition free energy as defined in equation (3). The free energy is calculated using the lowest Gibbs free configuration (standard conditions) at the $\omega$B97X-D3BJ/6-311++G(3df,3pd)//B97-3c level of theory. The x-axis shows the monomer added to the $(SA)_{10}(base)_{10}$ FNP where the label defines the base.

### 3.1.1 SA–DMA

From Figure 2 it is seen that the $(SA)_{10}(DMA)_{10}$ FNPs are the least favorable for uptake of vapors, having positive addition free energies for FA, IEPOX, and $\beta4$-ISOPOOH and higher than $-3$ kcal/mol addition free energies for the remaining monomers.

This is due to the several effects. First, SA–DMA FNPs are already substantially more stable than the other FNPs due to the high basicity of DMA compared to the other bases. (Wu et al., 2024) Hence, the reference system in equation (3) is already low in free energy, leading to higher addition free energies. Second, the two bulky methyl groups in the DMA molecules




will lead to some steric hindrance for adding the organics to the FNP. Third, adding the new organic monomer disrupts the already favorable hydrogen bond network of the SA–DMA FNPs, either destabilizing the system or only making it slightly

more favorable. This concept is illustrated in Figure 3 for the $(SA)_{10}(DMA)_{10}(trans\text{-}\beta\text{-}IEPOX)_1$ structure. The addition of the IEPOX molecule forms an epoxide–DMA bond (light-blue circle) which forces the methyl group on the DMA molecule slightly inside the cluster, which leads to increased steric repulsion. This is in contrast to the preferred orientation of the DMA molecule where both methyl groups point out of the cluster (dark blue square).

$$(SA)_{10}(DMA)_{10}(trans\text{-}\beta\text{-}IEPOX)_1$$

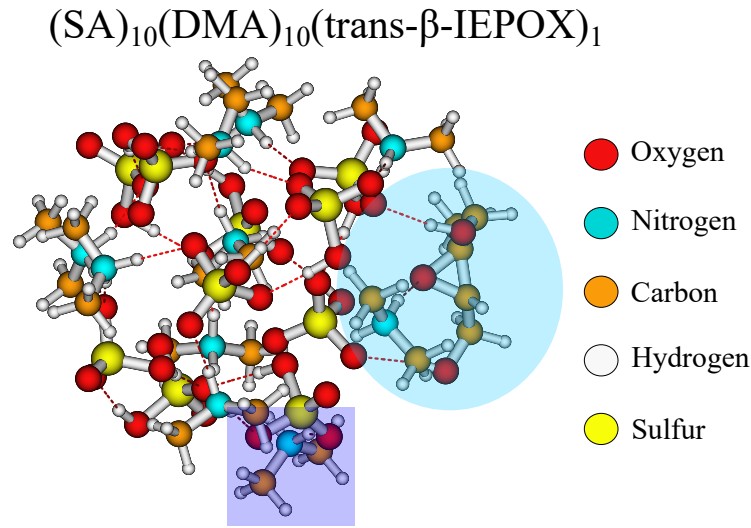

**Figure 3.** The $(SA)_{10}(DMA)_{10}(trans\text{-}\beta\text{-}IEPOX)_1$ cluster lowest in free energy (standard conditions) at the $\omega$B97X-D3BJ/6-311++G(3df,3pd)//B97-3c level of theory.

### 3.1.2   SA–TMA

For the $(SA)_{10}(TMA)_{10}$ FNPs all the addition free energies are negative, except for $\beta$1-ISOPOOH (+0.1 kcal/mol). This is mainly caused by the fact that the SA–TMA FNPs by themselves are relatively unstable. (Wu et al., 2024) Although TMA is a stronger base, in terms of gas-phase basicity, (Hunter and Lias, 1998) than DMA, its three bulky methyl groups and only a single hydrogen bond donor prevents it from obtaining a favorable hydrogen bond topological network. Hence, introducing

an additional monomer extends the hydrogen bond network and lowers the steric hindrance. It is curious that there is such a large difference between the two isomers of ISOPOOH (+0.1 vs −3.5 kcal/mol) as they structurally are very similar (see Figure 4). The geometries can not easily explain the difference as their carbon backbones point outwards from the cluster and do not interact with the "core". This leaves the main cluster geometry as the driving parameter. Here the $(SA)_{10}(TMA)_{10}(\beta 4\text{-}$ ISOPOOH$)_1$ cluster seems a bit less tightly packed compared to the $\beta$1-ISOPOOH cluster, which might equal a more favorable

hydrogen bonding topological network.



$(SA)_{10}(TMA)_{10}(\beta 1\text{-ISOPOOH})_1$ $(SA)_{10}(TMA)_{10}(\beta 4\text{-ISOPOOH})_1$

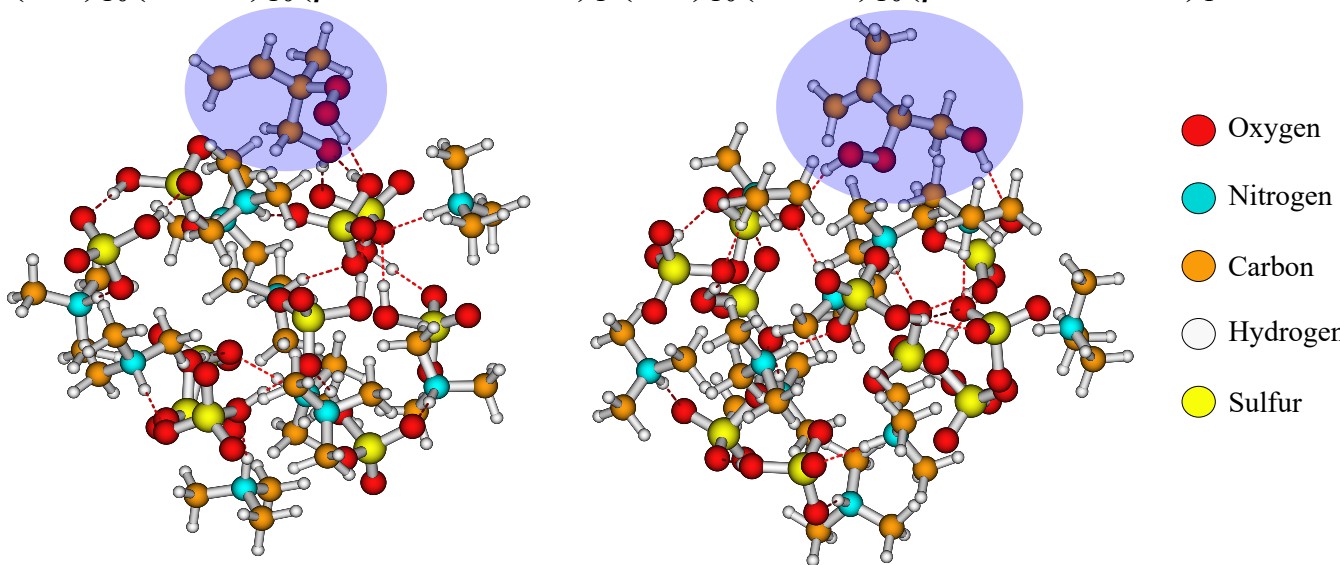

- ● Oxygen
- ● Nitrogen
- ● Carbon
- ○ Hydrogen
- ● Sulfur

**Figure 4.** The $(SA)_{10}(TMA)_{10}(\beta 1\text{-ISOPOOH})_1$ and $(SA)_{10}(TMA)_{10}(\beta 4\text{-ISOPOOH})_1$ cluster lowest in free energy (standard conditions) at the $\omega$B97X-D3BJ/6-311++G(3df,3pd)//B97-3c level of theory.

Likewise, it is also curious that the bulky HOM has addition free energy decrease as large ($-9.2$ kcal/mol) as the much smaller nitric acid ($-9.0$ kcal/mol) and pinic acid ($-6.9$ kcal/mol). Studying the structure (Figure 5), the HOM appears to compress the entire cluster "hovering" slightly above the surface favoring internal hydrogen bonding as seen on the top of the HOM. This means that as the first addition is very favorable the second addition of another HOM would probably also be likely as this would potentially compress the cluster even further. In addition, the internal hydrogen bonds in the HOM could potentially be broken to act as a tether between two adjacent clusters.

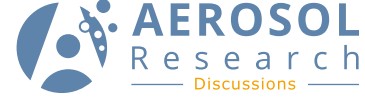

$(SA)_{10}(TMA)_{10}(HOM)_1$

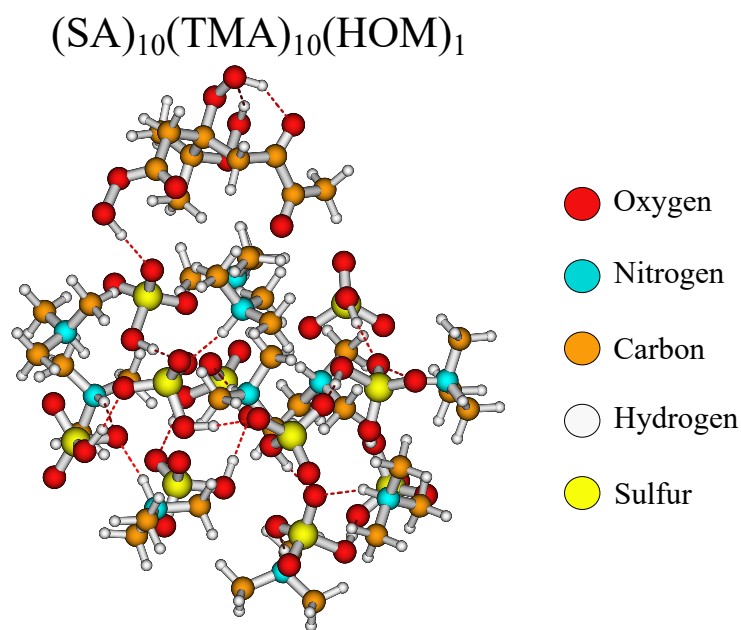

● Oxygen

● Nitrogen

● Carbon

○ Hydrogen

● Sulfur

**Figure 5.** The $(SA)_{10}(TMA)_{10}(HOM)_1$ cluster lowest in free energy (standard conditions) at the $\omega$B97X-D3BJ/6-311++G(3df,3pd)//B97-3c level of theory.

### 3.1.3 SA–MA

The SA–MA FNPs has a large addition free energy decrease for the HOM ($-9.6$ kcal/mol), pinic ($-9.2$ kcal/mol) and pinonic acid ($-9.7$ kcal/mol). It is interesting that they are almost equivalent, as pinic acid has two carboxylic acid, whereas in pinonic acid one of the carboxylic acid groups is exchanged for a carbonyl group and we would expect carboxylic acid to bind stronger.



$(SA)_{10}(MA)_{10}(Pinic)_1$ $\qquad$ $(SA)_{10}(MA)_{10}(Pinonic)_1$

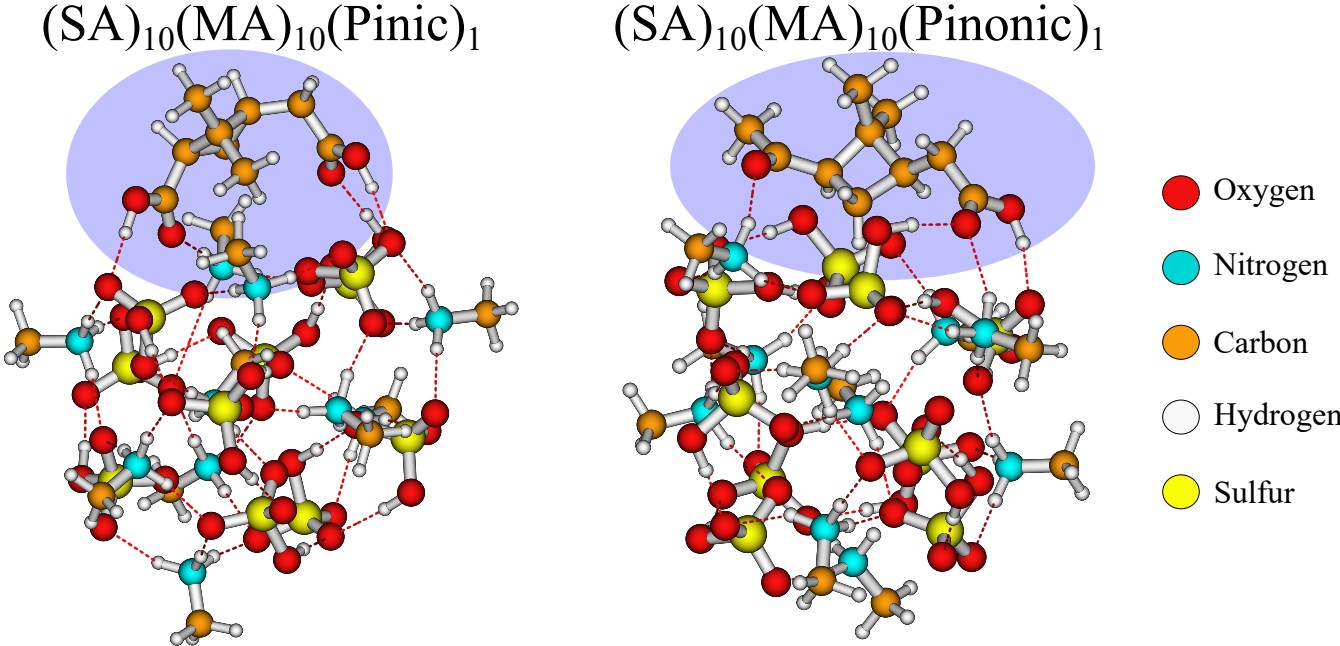

**Figure 6.** The $(SA)_{10}(MA)_{10}(Pinic)_1$ and $(SA)_{10}(MA)_{10}(Pinonic)_1$ cluster lowest in free energy (standard conditions) at the $\omega$B97X-D3BJ/6-311++G(3df,3pd)//B97-3c level of theory.

Studying the structures (Figure 6) we see they reside at the surface of the cluster interacting with their carboxylic acid/carbonyl group in almost the same way. This suggests the clusters can rearrange to accommodate both types of interactions and it does not matter much that one of the carboxylic acid groups has been exchanged with a carbonyl group.



$(SA)_{10}(MA)_{10}(HOM)_1$

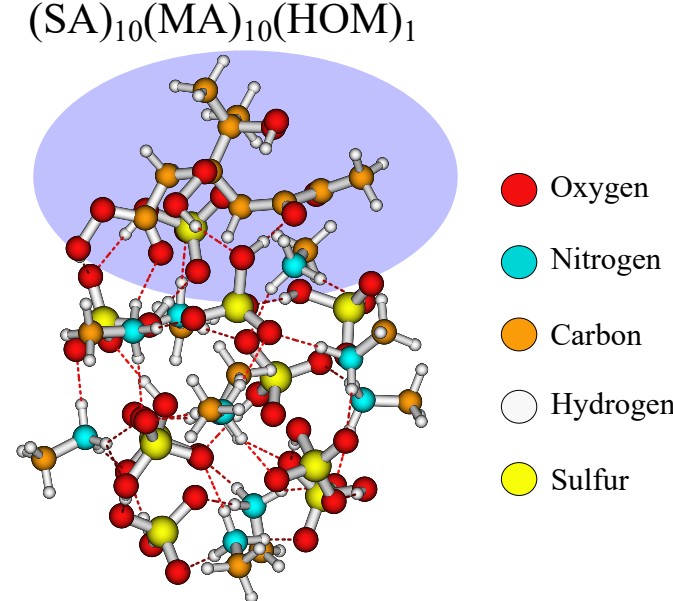

● Oxygen

● Nitrogen

● Carbon

○ Hydrogen

● Sulfur

**Figure 7.** The $(SA)_{10}(MA)_{10}(HOM)_1$ cluster lowest in free energy (standard conditions) at the $\omega$B97X-D3BJ/6-311++G(3df,3pd)//B97-3c level of theory.

Like the SA–TMA cluster it is quite favorable to add the HOM ($-9.6$ kcal/mol) but unlike the SA–TMA structure, this matches chemical intuition as the HOM almost maximizes the interactions with the cluster as seen in Figure 7. This also fits with the interaction pattern of pinic and pinonic acid explaining the addition free energy decrease.

### 3.1.4 SA–AM

The SA–AM FNPs have a similar large addition free energy decrease for pinic ($-8.4$ kcal/mol) and pinonic acid ($-10.5$ kcal/mol) as the SA–AM FNP. They also have the same structural characteristics. However, there is an inverse trend for the remaining monomers (FA, HOM, trans-$\beta$-IEPOX, $\beta$4-ISOPOOH, $\beta$1-ISOPOOH, and NA), where if the addition free energy decrease is large for one type of FNP, it is small for the other type of FNP for a given monomer. This is especially noticeable for $\beta$4-ISOPOOH monomer where the addition free energy is almost zero for the SA–AM FNP but $\approx -7$ kcal/mol for the SA–MA FNP. Structurally, it is unclear why the addition free energy differs so much as both interact with the cluster's outer layer, and neither of their "backbone" carbon chains interacts with the cluster (Figure 8, ISOPOOH has been marked with a dark blue box). The only major difference is that in the SA–AM FNP the dihedral angle between the two functional groups in ISOPOOH allows it to attach entirely to the surface, while for the SA–MA FNP the dihedral angle between the function groups causes it to be slightly more embedded in the cluster.



$(SA)_{10}(MA)_{10}(\beta4\text{-ISOPOOH})_1$    $(SA)_{10}(AM)_{10}(\beta4\text{-ISOPOOH})_1$

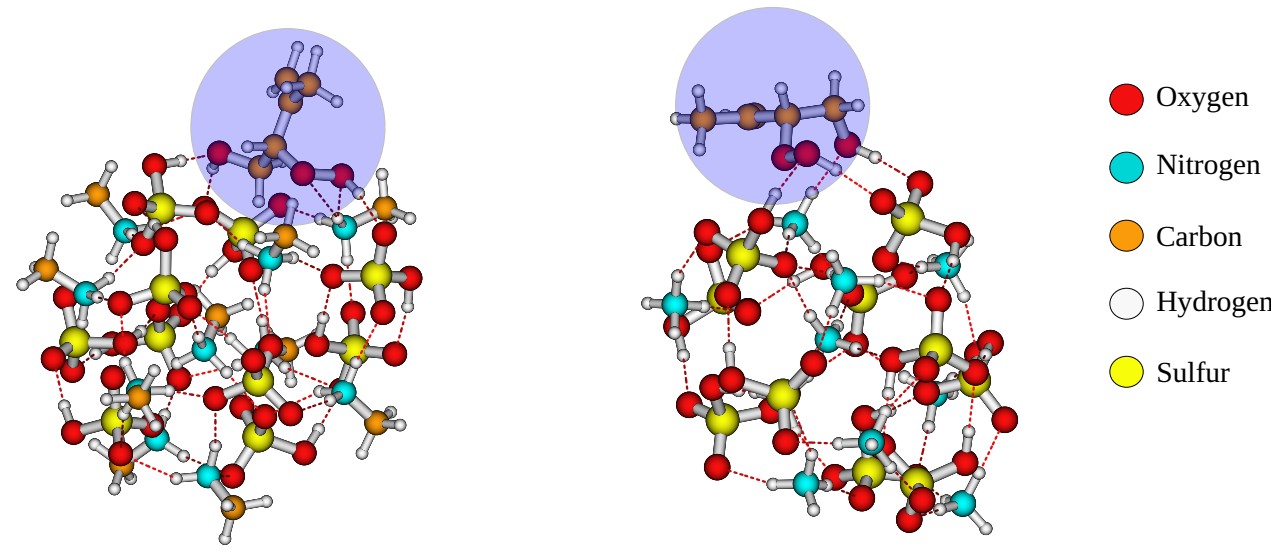

- 🔴 Oxygen
- 🔵 Nitrogen
- 🟠 Carbon
- ⚪ Hydrogen
- 🟡 Sulfur

**Figure 8.** The $(SA)_{10}(MA)_{10}\beta4\text{-ISPOOH})_1$ and $(SA)_{10}(AM)_{10}\beta4\text{-ISPOOH})_1$ cluster lowest in free energy (standard conditions) at the $\omega$B97X-D3BJ/6-311++G(3df,3pd)//B97-3c level of theory.

### 3.2 Potential for Organic Growth

It is clear that FNPs containing 10 acid–base pairs already have the potential to grow via organic uptake, as we observed addition
free energy exceeding $-8$ kcal/mol for the HOM, pinic acid, and pinonic acid. It appears that isoprene oxidation products have
a lower propensity for contributing to the early growth, compared to the $\alpha$-pinene products. This is in accordance with the
expected vapor pressure of the compounds.

Out of the studied organics, the HOM, pinic, and pinonic acid seem to be the most potent candidates for organic growth. To
further investigate this, we calculated the addition free energy of a second pinic and pinonic acid molecule. We chose to study
the SA–AM FNPs as they showed the largest potential for uptake of these species.

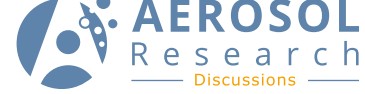

$(SA)_{10}(AM)_{10}(Pinic)_2$  $(SA)_{10}(AM)_{10}(Pinonic)_2$

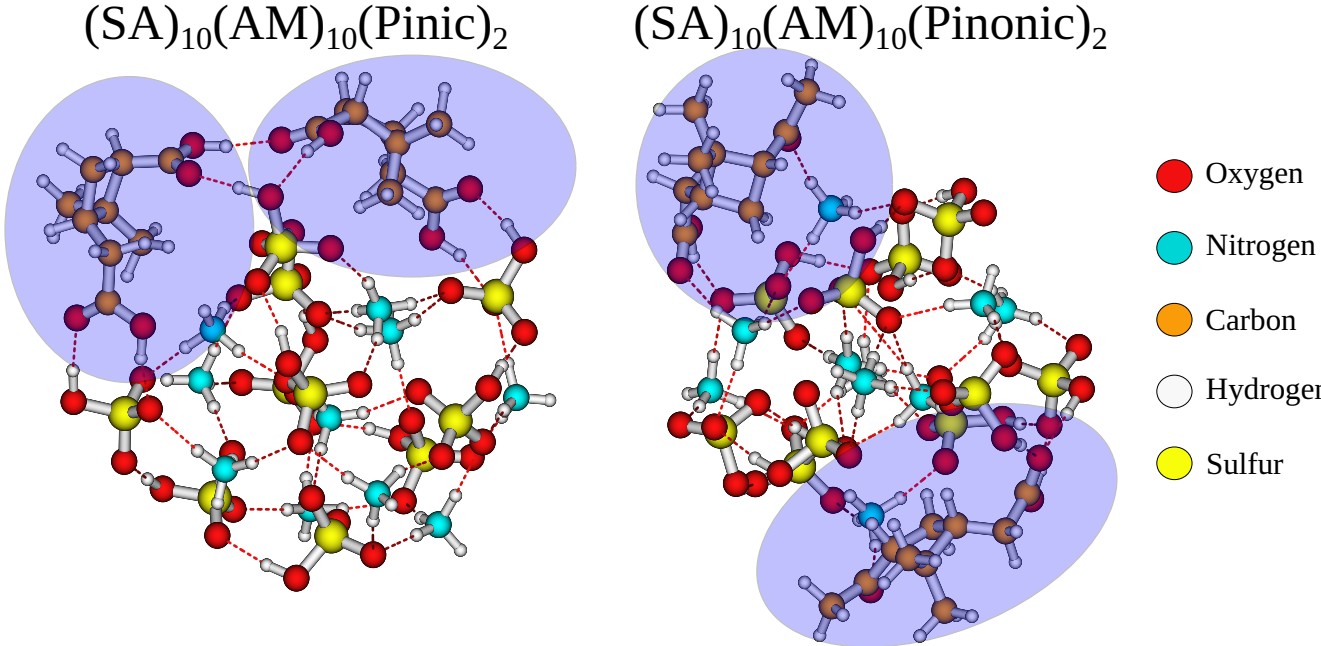

**Figure 9.** The $(SA)_{10}(AM)_{10}(Pinic)_2$ and $(SA)_{10}(AM)_{10}(Pinonic)_2$ cluster lowest in free energy (standard conditions) at the $\omega$B97X-D3BJ/6-311++G(3df,3pd)//B97-3c level of theory.

The addition free energy of the second addition is $-11.8$ and $-3.9$ kcal/mol for pinic and pinonic acid, respectively. The addition free energy decrease of pinic acid is even larger than the first ($-8.4$ kcal/mol). On the contrary, pinonic acid increased dramatically from the first addition ($-10.5$ kcal/mol). This stark contrast can easily be seen from the geometries in Figure

9. Pinic acid with its two carboxylic acid groups cannot only bind with the surface of the FNP but also with the other pinic acid. One could envision that with further additions of pinic acid molecules, they could link together over the entire surface essentially leading to a core-shell structure. In contrast, pinonic acid only has one carboxylic acid group which attaches to the surface, however, it does not link the two monomers together as the carbonyl groups favors binding with the bases in the FNP. This is in agreement with the earlier findings by Elm et al. (2017) and the cluster-of-functional groups approach by Pedersen

et al. (2024) which states that carboxylic acid groups yield the largest addition free energy decrease for organic enhanced atmospheric cluster formation. The fact that the organic–organic interaction is affecting the FNP growth is an intriguing result. This could indicate that uptake of organics on small particles, and in extension the growth thermodynamics, significantly deviate from the usually employed metric of the saturation vapor pressure of the organics. Hence, uptake of organics might be very dependent on simultaneously the organic–organic interactions and the organic–FNP interactions. This implies that

co-condensation of various vapors on FNPs will be very dependent on the exact functional groups in the molecules and should be futher studied in the future.



## 3.3 Free Energies Under Actual Conditions

The free energies calculated in the previous sections are at standard conditions ($p = 1$ atm, $T = 298.15$ K). As none of the involved species are present in such a high concentration in the atmosphere, the sign of the addition standard free energy does not tell if the addition is spontaneous in the atmosphere. Using Equation (5), the free energy can be calculated at the actual conditions. We have chosen to scan the free energy dependence on temperature and concentration for the pinic additions onto the $(SA)_{10}(AM/MA)_{10}$ FNPs, as it showed favorable first (and second for AM) standard addition free energy. We chose two concentration regimes for the FNP compositions in accordance with the *Clusteromics* series of papers. (Elm, 2021a, b, 2022; Knattrup and Elm, 2022; Ayoubi et al., 2023) In both the upper and lower regimes, SA is set to $10^6$ molecules cm$^{-3}$, the concentrations of bases were AM (10 ppt, 10 ppb), MA (1, 100 ppt) and DMA/TMA (1, 10 ppt) for the lower and upper regimes, respectively. Ambient measurements of pinic acid concentrations are hard to come by so we chose a range from 0.5 to 20.5 ppt as a representative concentration range.

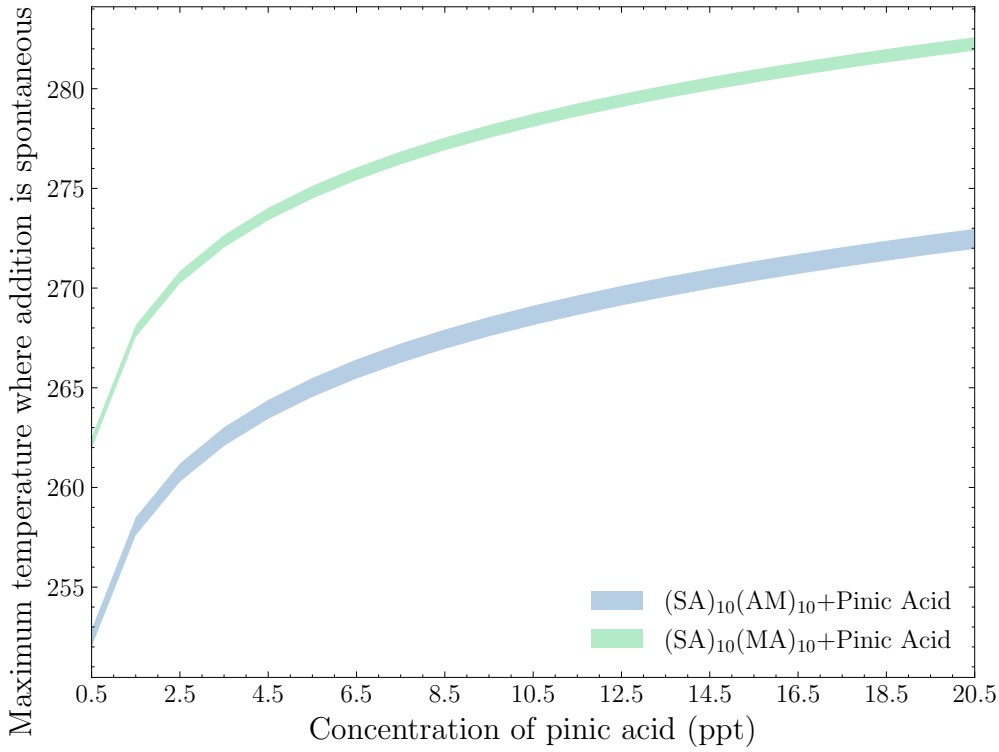

**Figure 10.** The maximum temperature where the addition of pinic acid onto either $(SA)_{10}(AM)_{10}$ or $(SA)_{10}(MA)_{10}$ is spontaneous at the given conditions ranging from the upper to the lower concentration regime (Top of distribution = upper concentration regime, bottom of distribution = lower concentration regime).





Figure 7 shows the maximum temperature (Equation (7)) where the addition of pinic acid onto either $(SA)_{10}(AM)_{10}$ or
$(SA)_{10}(MA)_{10}$ is spontaneous at the given conditions ranging from the upper to the lower concentration regime.

In general, the main factor affecting the spontaneity of the addition is the concentration of pinic acid and not the concentration of the components of the FNP. Changing the concentration of pinic acid from 0.5 ppt to 20 ppt changes the maximum spontaneous condensation temperature from 252 K to 273 K for the addition onto SA–AM FNPs and from 262 K to 282 K for the addition onto the SA–MA FNPs. Changing the concentration from the low to high limit of the bases yield at most a change
of 1 K. From the results, it is clear that when the concentrations are low (around 1 ppt) very cold temperatures (sub 252 K) are required, however, around 10 ppt the required temperature is around 270 K which is not unreasonable for atmospheric conditions. The addition onto the MA FNP is much more favorable as the maximum temperature is directly proportional to the addition free energy, for our concentration ranges, the roughly 0.8 kcal/mol difference in the addition free energies yields a temperature difference of 10 K.

The NA and FA addition onto the $(SA)_{10}(AM)_{10}$ FNP stands out compared to the other monomers, as the high concentrations of these species (ppb compared to ppt) may push the uptake to be spontaneous under actual conditions even though their standard addition free energy is above $-4$ kcal/mol. However, calculating the actual addition free energies at 20.5 ppb and in the high concentration regime, a temperature below 252 K and 239 K is still required for spontaneous addition of FA and NA, respectively. This is consistent with the CLOUD chamber experimental findings by Wang et al. (2020, 2022), that cold
temperatures corresponding to upper tropospheric conditions are required for NA to condense onto the particles. From equation (7) it is also clear that the log dependence on the concentrations minimizes the enhancement in temperature of going from ppt to ppb and that the addition free energy has the largest impact as it is directly proportional to the temperature where the addition becomes spontaneous.

## 4    Conclusions

We have investigated the ability of $(SA)_{10}(base)_{10}$ freshly nucleated particles (FNPs) with SA = sulfuric acid and base = ammonia (AM), methylamine (MA), dimethylamine (DMA), and trimethylamine (TMA) to uptake first-generation oxidation products of isoprene (trans-$\beta$-IEPOX, $\beta$4-ISPOOH, and $\beta$1-ISOPOH), $\alpha$-pinene (pinic and pinonic acid), a potential highly oxidized molecule (HOM), formic acid (FA), and nitric acid (NA). This was done using quantum chemical methods at the $\omega$B97X-D3BJ/6-311++G(3df,3pd)//B97-3c level of theory.

We find that the HOM, pinic, and pinonic acid can exhibit large decreases in addition free energies between $-8$ to $-10$ kcal/mol making them potential candidates for the organic growth of FNPs. This suggests that the studied isoprene oxidation products do not contribute to the early growth of FNPs, but the $\alpha$-pinene products do. To further investigate this we calculated the second addition free energy of pinic and pinonic acid onto the $(SA)_{10}(AM)_{10}$ FNP. We find that the pinic acid still exhibits a large addition free energy decrease of $-11.8$ kcal/mol but the secondary addition of pinonic acid drops to $-3.9$ kcal/mol.
The reason pinic acid maintains its high addition free energy decrease is due to its two carboxylic acid groups. The functional



groups enable the pinic acid monomer to bind not only to the FNP surface but also to adjacent pinic acid. The high potential for pinic addition is confirmed by calculating the addition free energy at atmospheric conditions.

In the future, we imagine molecular dynamics-based sampling techniques, such as umbrella sampling, can give better insight into the effects of multiple minima and partitioning of the uptake or collision simulations using molecular dynamics for realistic

collision coefficients.

*Data availability.*  All the calculated structures and thermochemistry are available in the Atmospheric Cluster Database (ACDB) (**?**Kubečka et al., 2023a)

*Author contributions.*  Conceptualization: J.E.;

Methodology: Y.K., J.E.;

Formal analysis: Y.K.;

Investigation: Y.K.;

Resources: J.E.;

Writing - original draft: Y.K.;

Writing - review & editing: Y.K., J.E.;

Visualization: Y.K.;

Project administration: J.E.;

Funding acquisition: J.E;

Supervision: J.E.

*Competing interests.*  At least one of the (co-)authors is a member of the editorial board of Aerosol Research. The authors have no other

competing interests to declare.

*Acknowledgements.*  Funded by the European Union (ERC, ExploreFNP, project 101040353). Views and opinions expressed are however those of the authors only and do not necessarily reflect those of the European Union or the European Research Council Executive Agency. Neither the European Union nor the granting authority can be held responsible for them.

This work was funded by the Danish National Research Foundation (DNRF172) through the Center of Excellence for Chemistry of

Clouds.

The numerical results presented in this work were obtained at the Centre for Scientific Computing, Aarhus https://phys.au.dk/forskning/ faciliteter/cscaa/.





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
