# Peer review of "Uptake of Organic Vapors and Nitric Acid on Atmospheric Freshly Nucleated Particles"

_Aerosol Research, 2024_

## Author Comment (AC1)

**Response to Reviews**

We highly appreciate the positive comments from both the reviewers and all the points have been addressed in the revised paper. We hope that the following responses are satisfying and that the paper can be accepted for publication. The reviewers' comments have been reproduced in blue text below, followed by our point-by-point replies.

**Adressing an error**

Since submitting this paper, we have discovered a bug in the program that calculates the binding free energies under given conditions. The concentrations were a factor 1000 too high, i.e., ppt was incorrectly set to ppb, and ppb was incorrectly set to ppm. However, as the concentrations enter a logarithm, the consequent free energy change is a relative constant offset by $(\ln(1000) = 6.9)$, which made this error extremely difficult to catch. This bug only affects the results in section 3.3 and a sentence in the conclusion and abstract. All other sections remain the same, except for the changes warranted by the reviewers.

We have redone the calculations for the section and adjusted the section accordingly. In short, the main conclusion of the section is now that the temperature required for spontaneous uptake of the monomer products is too low at ppt concentrations to be realistic. We speculate that this is because the FNPs are too small, and the uptake monomers cannot overcome the Kelvin effect. Therefore, we have supplemented the section with an extra calculation of an accretion product ("dimer") that has more interaction groups. We show that it can have spontaneous uptake under realistic conditions. We also removed the last equation in the theory section as it was never actually used in the manuscript.

We sincerely apologize for this error, but believe that in the process, this actually helped improve the manuscript as it prompted us to include a calculation using a very large organic dimer. This is the first time such a large calculation has been carried out at the DFT level.

**Anonymous Referee #1**

Knattrup and Elm investigated the potential uptake of first-generation oxidation products from atmospherically relevant VOCs on (SA)(base) freshly nucleated clusters using DFT and highlight that dicarboxylic acids could potentially contribute to the growth of 2nm freshly nucleated particles. This study is important for understanding the cluster-to-particle transition process in sulfuric-based clusters. The manuscript is very well written, and results are of substantial relevance to Aerosol science. Therefore, I recommend publication to Aerosol Research after the following minor comments have been addressed.

Line 97: Do the authors mean 298.15 K?

**Author reply:**
Yes, it has been changed accordingly.

**Line 98, Page 4**

From: 298.5

To: 298.15

The punctuation needs to be checked at some places in the manuscript

**Author reply:**
We have given the paper another thorough read and tried to remove formatting errors, grammatical issues, and unclear language.

Most calculations have shown that the binding strength of ammonia and amines to sulfuric acid-based clusters is generally in this order: DMA>MA>$NH_3$. How do you explain that this is not the case when most first-generation oxidation products are involved (see Figure 2)?

**Author reply:**
The values displayed in the figure are addition free energies, i.e., the binding free energy of the clusters themselves is set as the zero-point which means we lose the information about the ordering. The binding free energies of the FNP follow MA>DMA>AM>TMA. The DMA>MA>$NH_3$ order only holds for smaller clusters, where the acid–base strength interaction dominates. However, for larger clusters, the sterical hindrance becomes important, and the binding strength no longer just follows the acid–base strengths. We agree that this is unclear from the manuscript and have added a sentence to address it.

**Line 151, Page 6**

Added: It should be noted that the most negative binding free energy of the FNPs follows the order: MA > DMA > AM > TMA. This trend does not align with the gas-phase acidity or basicity of the components. Instead, steric factors become increasingly important for larger sizes, changing the order.

Beside the free energy of addition, what do the authors think about the atmospheric concentrations of first-generation oxidation products used in this study? Are their concentrations high enough to cause effective changes in the cluster-to-particle transition process in the actual atmospheric environments?

**Author reply:**
With the new conclusion in Section 3.3, we do not believe the first-generation oxidation products will affect the cluster-to-particle transition process, as this would require several ppb or higher concentrations and low temperatures. However, we believe it is an interesting perspective that the organics might change the cluster-to-particle transition point. We have added a sentence as an outlook in the conclusion.

**Line 3, Page 18**

Added: It would also be interesting to study whether organic compounds can influence the cluster-to-particle transition points.

**Anonymous Referee #2**

Knattrup and Elm conducted simulations to investigate the potential uptake of sulfuric acid, ammonia, and amines on first-generation oxidation products from common SOA gas precursors. This study is significant for understanding the reaction mechanism of the second addition of sulfuric acid, ammonia, and ammonia on SOA. The work was well-designed, and the paper was well-organized. The results strongly support the authors' conclusions. Overall, I recommend this paper to be published. However, I have a few minor comments I want the authors to consider.

Could you comment on how RH and higher temperature (e.g., PBL condition) will affect your results?

**Author reply:**
We agree that the effects of relative humidity could be important, however, for cluster of these sizes, the effect would have to be explicitly calculated to be quantified. This would require significantly more calculations, as we would have to re-calculate each FNP with a varying number of explicit water molecules. Furthermore, for each of those, we would have to calculate the addition free energy. We are currently looking into the hydration of FNPs in a separate study.

Regarding the temperature, in the rewritten Section 3.3, we find that extremely cold temperatures are required for spontaneous uptake, this means that we would not expect organic monomer uptake on FNPs at this size at higher temperatures.

Could you explain why you chose (SA)10(AM)10? I am not very familiar with this.

**Author reply:**
These 10 acid - 10 base clusters were chosen as they surpass the cluster-to-particle transition point. This means the cluster exhibits more particle-like properties (solvated ions and a leveling out of the average binding free energy per monomer). We agree that our choice of size was unclear and have added the following sentence.

**Line 67, Page 3**

Added: We chose these sizes as the clusters have all reached the cluster-to-particle transition point.

It will be good to include the chemical reaction mechanism in the main manuscript to help readers understand the reaction.

**Author reply:**
We agree with the reviewer that it would be beneficial to add the chemical reaction for the uptake.

**Line 63, Page 3**

From: In this paper, we study the uptake of NA and common organics on the $(SA)_{10}(AM/MA/DMA/TMA)_{10}$ FNPs using quantum chemical methods.

To: In this paper, we study the uptake of NA and common organics (denoted X) on the $(SA)_{10}(AM/MA/DMA/TMA)_{10}$ FNPs using quantum chemical methods for the following reaction scheme:

$$(SA)_{10}(base)_{10} + (X)_1 \rightarrow (SA)_{10}(base)_{10}(X)_1 \tag{1}$$